# Real-time PCR detection of mixed *Plasmodium ovale curtisi* and *wallikeri* infections in human and mosquito hosts

**Varun R. Potlapalli**[1], **Meredith S. Muller**[1], **Billy Ngasala**[2], **Innocent Mbulli Ali**[3], **Yu Bin Na**[1], **Danielle R. Williams**[4], **Oksana Kharabora**[1], **Srijana Chhetri**[1], **Mei S. Liu**[1], **Kelly Carey-Ewend**[5], **Feng-Chang Lin**[5], **Derrick Mathias**[6], **Brian B. Tarimo**[7], **Jonathan J. Juliano**[1,5], **Jonathan B. Parr**[1], **Jessica T. Lin**[1] *

1 Institute of Global Health and Infectious Diseases, University of North Carolina School of Medicine, Chapel Hill, North Carolina, United States of America, 2 Muhimbili University of Health and Allied Sciences, Dar es Salaam, Tanzania, 3 Department of Biochemistry, Faculty of Science, University of Dschang, Dschang, Cameroon, 4 Department of Microbiology and Immunology, University of North Carolina, Chapel Hill, North Carolina, United States of America, 5 Gillings School of Global Public Health, University of North Carolina, Chapel Hill, North Carolina, United States of America, 6 Florida Medical Entomology Laboratory, Institute of Food & Agricultural Sciences, University of Florida, Vero Beach, Florida United States of America, 7 Vector Immunity and Transmission Biology Unit, Department of Environmental Health and Ecological Sciences, Ifakara Health Institute-Bagamoyo Office, Bagamoyo, Tanzania

* jessica_lin@med.unc.edu

**Data Availability Statement:** All relevant data are within the manuscript.

**Funding:** This work was supported by the National Institutes of Health through grants R01AI137395

## Abstract

*Plasmodium ovale curtisi* (*Poc*) and *Plasmodium ovale wallikeri* (*Pow*) represent distinct non-recombining *Plasmodium* species that are increasing in prevalence in sub-Saharan Africa. Though they circulate sympatrically, co-infection within human and mosquito hosts has rarely been described. Separate 18S rRNA real-time PCR assays that detect *Poc* and *Pow* were modified to allow species determination in parallel under identical cycling conditions. The lower limit of detection was 0.6 plasmid copies/μL (95% CI 0.4–1.6) for *Poc* and 4.5 plasmid copies/μL (95% CI 2.7–18) for *Pow*, or 0.1 and 0.8 parasites/μL, respectively, assuming 6 copies of 18s rRNA per genome. However, the assays showed cross-reactivity at concentrations greater than $10^3$ plasmid copies/μL (roughly 200 parasites/μL). Mock mixtures were used to establish criteria for classifying mixed *Poc/Pow* infections that prevented false-positive detection while maintaining sensitive detection of the minority ovale species down to $10^0$ copies/μL (<1 parasite/μL). When the modified real-time PCR assays were applied to field-collected blood samples from Tanzania and Cameroon, species identification by real-time PCR was concordant with nested PCR in 19 samples, but additionally detected two mixed *Poc/Pow* infections where nested PCR detected a single *Po* species. When real-time PCR was applied to oocyst-positive *Anopheles* midguts saved from mosquitoes fed on *P. ovale*-infected persons, mixed *Poc/Pow* infections were detected in 11/14 (79%). Based on these results, 8/9 *P. ovale* carriers transmitted both *P. ovale* species to mosquitoes, though both *Po* species could only be detected in the blood of two carriers. The described real-time PCR approach can be used to identify the natural occurrence of mixed *Poc/Pow* infections in human and mosquito hosts and reveals that such co-infections and co-transmission are likely more common than appreciated.

to JTL, R21AI148579 to JP and JTL, R21AI152260 to JTL, and K24AI134990 to JJJ. IMA was supported by a postdoctoral fellowship from the Wellcome Trust [grant # 107741/A/15/Z] and the UK Foreign, Commonwealth and Development Office, with support from the Developing Excellence in Leadership, Training and Science in Africa (DELTAS Africa) program. The funders had no role in the study design, data collection and analysis, decision to publish, or preparation of the manuscript.

**Competing interests:** The authors have declared that no competing interests exist.

## Author summary

*Plasmodium ovale*, one of five species of malaria known to infect humans, in fact represents two distinct species, *P. ovale curtisi* (*Poc*) and *wallikeri* (*Pow*), that can only be distinguished using molecular diagnostics. Though *Poc* and *Pow* circulate in the same regions in Africa and Asia, mixed infections, where both are found in the same human host, have rarely been described. In this study, we modified existing real-time PCR assays targeting 18S rRNA and developed an algorithm to detect mixed *Poc*/*Pow* infections. We then applied these assays to field-collected samples from Tanzania and Cameroon, including blood samples from *P. ovale*-infected persons and *P. ovale*-positive mosquito midguts saved from mosquito feeding assays. We detected both *Poc* and *Pow* in roughly 10% of human *P. ovale* blood-stage infections, and surprisingly, in a majority of blood-fed mosquitoes. This suggests that *Poc* and *Pow* co-infect the same hosts more frequently than previously realized.

## Introduction

*Plasmodium ovale curtisi* (*Poc*) and *Plasmodium ovale wallikeri* (*Pow*), long recognized simply as *Plasmodium ovale* due to their similar morphology under the microscope, in fact represent two distinct, non-recombining malaria species [1,2]. *Plasmodium ovale* (*Po*, inclusive of both species) has more commonly been reported from West Africa [3], but its prevalence based on recent PCR surveys appears to be increasing in East Africa [4–8]. There is some evidence that the two species differ in their latency period or relapse periodicity, as well as their presentation in travelers [3,9–12]. Still, the extent to which *Poc* and *Pow* differ in their biology, epidemiology, and clinical manifestations in Africa, where they are most frequently found, remains an active area of research [13,14].

Interestingly, though both *P. ovale* species circulate sympatrically in time and space, mixed infections of the two species have rarely been described, with <20 cases of *Poc*/*Pow* co-infection reported across >35 studies encompassing 1,515 *P. ovale* cases in the literature [3,13,15,16]. Since the existence of mixed *Plasmodium* infections of other species is well-documented, we suspected this was due to technical limitations rather than biological constraints. We adapted previously published nested and real-time PCR (qPCR) assays for *Po* species detection to establish criteria for defining when both *Poc* and *Pow* are present, and then applied these assays to *Po*-positive isolates from Africa. Our methods and findings add to the growing body of work on better defining the epidemiology of these two species in Africa.

## Methods

### Ethics statement

This study was approved by institutional review boards at the University of North Carolina (IRB #16–0079), Tanzania National Institute for Medical Research (TransMIT 3639/Vol 34/005), Muhimbili University of Health and Allied Sciences (MUHAS-REC-5-2021-124), Ifakara Health Institute (IHI/IRB/AMM/ No:12–2020), and the Cameroon Baptist Convention Health Board (IRB2019-40). Written informed consent was obtained from all participants.

Previously published molecular assays for the detection and differentiation of *P. ovale* species were reviewed (**Table 1**). Based on initial testing, a nested PCR (nPCR) assay from Calderaro, et. al [17,18] and real-time qPCR assays from Perandin, et al. [19] and Calderaro, et. al

**Table 1. Published molecular assays to distinguish *P. ovale* species, *Po curtisi* (*Poc*) and *Po wallikeri* (*Pow*), and their application to clinical samples.**

| | Target | Method details (Primers/probes) | Country | P. ovale | Poc | Pow | Poc/Pow mixed | Comments |
|---|---|---|---|---|---|---|---|---|
| **Nested PCR** | | | | | | | | |
| **Calderaro, et. al.** [17] J Clin Micro, 2007 | **18S rRNA** (~800bp) | rPLU1, rPLU5 (*Plasmodium*) rOVA1, rOVA2 (*Poc*) rOVA1v, rOVA2v (*Pow*) | Return travelers to Italy (n = 62) | 10 | Not specified | | | |
| **Oguike, et al.** [22] Int J Parasitol, 2011 | **PoTRA** (tryptophan-rich antigen) | PoTRA3 fwd, PoTRA3 rev (44 cycles) PoTRA5 fwd, PoTRA5 rev (44 cycles) 245 bp product (*Pow*) 317–335 bp product (*Poc*) | Congo (n = 6) Equatorial Guinea (n = 49) Uganda (n = 1,254) | 6 4 36 | 2 2 16 | 4 2 20 | 0 0 1* | *1 mixed Poc/Pow not detected by PoTRA assay, but confirmed with *pog3p* sequence analysis. |
| **Tanomsing, et al.** [23] J Clin Micro, 2013 | **PoTRA** | PoTRA-F, PoTRA3 rev (25 cycles) PoTRA-F + PocTRA-R (30 cycles) 299 bp or 317 bp product (*Poc*) PoTRA-F + PowTRA-R (30 cycles) 245 bp product (*Pow*) | Not specified (n = 17) | 17 | 7 | 10 | 0 | Detection sensitivity: 2 to 10 parasites/µL blood. |
| **Real-time qPCR** | | | | | | | | |
| **Perandin, et al.** [19] J Clin Micro 2004 | **18S rRNA** | OVA-F, OVA-R Ova (VIC) probe (45 cycles, 60˚C annealing) | Returning travelers to Italy (n = 61) | 3 | 3* | 0 | 0 | Detects *Po curtisi* *Ranging 360–25000 parasites/uL |
| **Oguike, et al.** [22] Int J Parasitol, 2011 | **Porbp2** (Reticulocyte-binding protein homologue) | Porbp2TMfwd, Porbp2TMrev Sybr Green (40 cycles) 73˚C melt peak (*Poc*) 74˚C melt peak (*Pow*) | Equatorial Guinea (n = 49) Uganda (n = 1254) | 4 36 | 2 16 | 2 20 | 0 0 | Melting curve analysis assay |
| **Calderaro, et al.** [20] PLoS One 2012 | **18S rRNA** | OVA-Fv primer, OVA-Rv primer Ovav (FAM) probe (55 cycles, 60˚C annealing) | Not specified (n = 31) | 31 | 20 | 11 | 0 | Detects *Po wallikeri* Ct 54 threshold for positivity Parasitemia range: 50 to 20,500 parasites/uL Detection sensitivity of 50 plasmid copies/uL |
| **Nijhuis, et al.** [24] Eur J Clin Microbiol Infect Dis, 2018 | **18S rRNA** | *P. ovale curtisi-s* forward *P. ovale curtisi-TXR-MGB* probe *P. ovale wallikeri-s* forward *P. ovale wallikeri-TXR-MGB* probe *P. generic-as* reverse (45 cycles) | Not specified (n = 56) | 7 | Not specified | | | Multiplex real-time qPCR assay |
| **Lamien-Meda, et al.** [25] Acta Tropica, 2019 | **Clpc** (Caseinolytic protease C) | For qPCR: POvaDif F, POvaDif R EvaGreen dye Optional nested snapback forward primer for additional qPCR (50 cycles) For High-Resolution Melting: *Pow* Tm: 71.04˚C *Poc* Tm: 71.26˚C | Bangladesh and Ethiopia (n = 33) | 33 | 14 | 19 | 0 | *P. ovale curtisi and P. ovale wallikeri able to be differentiated with a small ΔTm (0.2˚C) 88% specificity for *P. ovale* spp. |
| **Amplicon Deep Sequencing** | | | | | | | | |
| **Mitchell, et al.** [26] J Infect Dis, 2021 | **18S rRNA** | Two round amplification; first using *Plasmodium* primers (20 cycles), then primers to ligate bar-coded Illumina adapters (40 cycles) SeekDeep (Version 3.0.1-dev) used to identify Po species based on sequence alignment | Democratic Republic of Congo (n = 62) | 62 | 23 | 16 | 3 | 20 samples could not be speciated due to filtering cutoffs for quality assurance. Samples that were successfully sequenced were of higher parasitemia (median density 23 vs 9 parasites/uL, p = 0.007) |

[20,21], all targeting the small subunit RNA gene (18S rRNA), were selected for further assay development and comparison.

## Modifications to the nested PCR assay

The nPCR assay published by Calderaro, et al.[17] was performed as described using HotStar-Taq (Qiagen). First round amplification of *Plasmodium ovale* inclusive of *Poc* and *Pow* used rPLU1 (TCAAAGATTAAGCCATGCAAGTGA) and rPLU5 (CCTGTTGTTGCCTTAAACTTC), resulting in a ~800 bp PCR product. Second round amplification for *Poc* or *Pow* species-specific detection was performed using rOVA1 (ATTTTGAAGAATACACTAGG) and rOVA2 for *P. ovale curtisi* (GGAAAAGGACACATTAATTGTATCCTAGTG), and rOVA1v (ATCTCTTTTGCTATTTTTTAGTATTGGAGA) and rOVA2v (GGAAAAGGACACTATAATGTATCCTAATA) for *P. ovale wallikeri*. This round resulted in a 787–789 and 782 bp product for *Poc* and *Pow*, respectively. A *Plasmodium ovale curtisi* 18S rRNA plasmid (MRA-180, BEI Resources) and genomic DNA from a sequence-confirmed *Pow* clinical sample from Tanzania were used as positive controls. When PCR products were not visualized on the 1% agarose gel using the standard conditions, the nPCR was repeated with adjustments made to cycle number, annealing temperatures, and input DNA volume to increase yield. The number of cycles was increased from 35 cycles in both rounds (70 total cycles) to as high as 40 cycles in round 1 and 45 cycles in round 2 (85 total cycles). Annealing temperatures were dropped in the second round species-specific PCR from 60°C and 58°C for *Poc* and *Pow*, respectively, to as low as 58°C and 56°C, respectively. Finally, the input DNA volume was increased from 5 μL up to 10 μL for the round 1 PCR, but maintained at 5 μL for the round 2 PCR. A subset of nPCR products underwent Sanger sequencing for confirmation of species designations.

## Modifications to real-time PCR assays

The sensitivity of the real-time PCR assays designed by Perandin, et. al [18] and Calderaro, et. al [19] to detect *Poc* and *Pow* was tested using dilutions of plasmids containing the small sub-unit ribosomal RNA gene (18S rRNA) specific to each species. The *Poc* 18S rRNA plasmid was obtained from BEI resources (MRA-180; GenBank: AF145337, 1,100 bp insert). The *Pow* plasmid control was created by cloning the first round nested PCR product of a Sanger-sequenced *Pow* clinical sample from Tanzania (MqTZ-0123) into a Topo 2.1kb vector using One Shot Top10 competent cells (Thermofisher Scientific). Real-time PCR was carried out using FastStart Universal Probe Master mix (ROX, Roche) and published primer and probe concentrations on a Bio-Rad CFX Connect Real-Time PCR Detection System.

To maximize detection sensitivity of both species, both assays were run in parallel to 50 cycles, instead of the originally published 45 and 55 cycles. For *P. ovale curtisi* amplification, OVA-F (TTTTGAAGAATACATTAGGGATACAATTAATG) and OVA-R (CATCGTTCCTCTAAGAAGCTTTACAAT) were used along with OVA (VIC) probe (CCTTTTCCCTATTCTACTTAATTCGCAATTCATG). For *P. ovale wallikeri* amplification, OVA-Fv (TTTTGAAGAATATATTAGGATACATTATAG) and OVA-R were used along with Ovav (FAM) probe (CCTTTTCCCTTTTCTACTTAATTCGCTATTATG). Product sizes for *Poc* and *Pow* were 127 and 130 bp, respectively. A common annealing temperature of 52.8°C was chosen for yielding similar Ct thresholds for detection of the same plasmid copy concentrations. At this lower annealing temperature, the assays could not be multiplexed into a duplex assay without detecting both species in each run regardless of the species-specific plasmid used, likely owing to the two real-time PCR assays having identical reverse primers and forward primers that differ by only two nucleotides, in addition to probes that also differ by only two nucleotides.

Thus, species-specific assays were run side-by-side in separate reactions under the same conditions.

### Real-time PCR of mono- and mixed species samples

The sensitivity of the modified qPCR assays for their respective 18S targets was tested for plasmid concentrations ranging from $10^5$ down to $10^{-1}$ plasmid copies/$\mu$L. A limit of detection was calculated for each assay using a Probit analysis [27]. To determine how the assays would perform for detecting samples with both species present, mock plasmid DNA control mixtures were created with *Poc* and *Pow* concentrations at a lower range ($10^2$ to $10^{-1}$ plasmid copies/$\mu$L) in ratios of 1:1, 1:2, 1:5, and 1:10 to simulate clinical samples. Each mixture was run 10 times using the qPCR assays, with the mean Ct value from the 10 runs reported. Based on these data, a classification scheme was developed for identifying mixed species infections (*Poc* and *Pow*).

### *P. ovale* mixed species detection in clinical blood and mosquito samples

DNA extracted from blood samples from Cameroon (n = 16) and Tanzania (n = 22) previously identified as *P. ovale*-positive based on an 18S qPCR that detects both *P. ovale* species [26] were used to compare the performance of the selected nPCR and modified qPCR assays. The Cameroon study was a prospective hospital based cross-sectional survey in the three main health facilities in the Dschang Health District in the West region of Cameroon that enrolled 431 patients between June-September 2020 that underwent screening for malaria per the attending physician [28]. Tanzanian samples were drawn from symptomatic children screened for a malaria therapeutic efficacy study at Yombo Clinic in Bagamoyo in 2016–2017 (YB- samples, [29]) as well as asymptomatic children and adults undergoing screening for a malaria transmission study from 2018–2019 [MqTZ- samples [8,30]]. Samples with lower Po 18S Ct values were deliberately chosen for study. The proportion of samples that amplified in each assay (nPCR and modified qPCR) and assay concordance for *P. ovale* species identification were examined, including whether there was detection of mixed species infection.

Additionally, real-time PCR detection of *P. ovale* species was performed on DNA extracted from 17 *P. ovale*-positive mosquito midgut samples. These were obtained from mosquito feeding assays performed on *P. ovale*-carriers in Tanzania using colony-reared *Anopheles gambiae* IFAKARA strain [8], including both direct skin feeding and membrane feeding assays. Mosquito midguts that were dissected and oocyst-positive by microscopy at day 8 post-blood feeding were stored in either ethanol or DNA/RNA shield (Zymo Research) then subjected to DNAzol-based DNA extraction (Invitrogen). Extracted midgut DNA was amplified using *Plasmodium* genus-specific primers in a conventional PCR as a first round reaction [31]. A second round *P. ovale* qPCR was performed on a 1:50 dilution of the first round product. Samples found to be positive were selected for further *P. ovale* species identification. Dried blood spot or leukodepleted blood samples obtained at the time of mosquito feeding were also subjected to real-time PCR species detection to compare the presence and type of *P. ovale* species detected in the blood vs. mosquito midgut samples.

## Results

### Real-time PCR detection of *P. ovale curtisi* and *wallikeri* in mono- and mixed infections

Run individually, both modified real-time PCR assays consistently detected down to $10^0$ 18S plasmid copies/$\mu$L of their respective *P. ovale* species, or the equivalent of roughly 0.2

**Table 2. Limit of detection and cross-reactivity of real-time PCR assays targeting *P. ovale curtisi* and *wallikeri*.** The mean Ct of the positive qPCR runs for each assay is shown for 18S plasmid concentrations ranging from $10^5$ to $10^{-1}$ copies/$\mu$L.

| *P. ovale curtisi* assay | | | *P. ovale wallikeri* assay | | |
|---|---|---|---|---|---|
| 18S plasmid copies/$\mu$L | Poc | Pow | 18S plasmid copies/$\mu$L | Poc | Pow |
| | No. positive qPCR runs (mean Ct) | | | No. positive qPCR runs (mean Ct) | |
| $10^5$ | 5/5 (27.4) | 2/5 (44.5) | $10^5$ | 5/5 (48.0) | 5/5 (26.9) |
| $10^4$ | 5/5 (31.9) | 2/5 (47.2) | $10^4$ | 4/5 (48.9) | 5/5 (30.8) |
| $10^3$ | 5/5 (36.7) | 2/5 (46.3) | $10^3$ | 1/5 (47.2) | 5/5 (35.1) |
| $10^2$ | 10/10 (38.1) | 0/10 | $10^2$ | 0/10 | 10/10 (39.4) |
| $10^1$ | 10/10 (42.3) | 0/10 | $10^1$ | 1/10 (49.5) | 10/10 (43.0) |
| $10^0$ | 10/10 (46.6) | 0/10 | $10^0$ | 1/10 (49.2) | 10/10 (48.4) |
| $10^{-1}$ | 5/10 (47.9) | 0/10 | $10^{-1}$ | 0/10 | 1/10 (49.2) |
| DI water | 8/29 (49.4) | | | 1/36 (48.9) | |

parasites/$\mu$L assuming 6 copies of 18S rRNA per genome (**Table 2**). Since distilled water negative controls also sometimes demonstrated late cycle amplification for the *Poc* assay (mean Ct = 49.4, range 49.2–49.9), Ct thresholds for positivity were set at Ct <49 for both species. Using these Ct thresholds, the limit of detection of the *Poc* assay based on Probit analysis was slightly lower than that of the *Pow* assay (0.6 plasmid copies/$\mu$L (95% CI 0.4–1.6) vs. 4.5 plasmid copies/$\mu$L (95% CI 2.7–17.7) for *Poc* and *Pow*, respectively). At higher concentrations of *Poc* and *Pow* plasmids, on the order of $10^3$ copies/$\mu$L, cross-reactivity between species was observed (**Table 2**).

The cross-reactivity between assays at higher parasite densities ($10^3$ copies/$\mu$L or >100–200 parasites/$\mu$L) could lead to misidentification of secondary species when none is present. The opposite may occur at lower parasite densities. To understand how the qPCR assays would perform for detecting mixed *P. ovale* species infections at lower densities, mock plasmid mixtures were created to mimic different *Poc* and *Pow* ratios of species at different concentrations (**Fig 1**). Both species remained detectable in the 1:1 mixtures down to their previous limits of detection. In fact, the sensitivity of the *Pow* assay appears "boosted" in the presence of *Poc* at the lowest concentration ($10^{-1}$ copies/$\mu$L), likely due to low-level cross-species reactivity though none was observed at this level in the mono-species reactions. In the 1:2 and 1:5 mixtures, detection of the minor species was preserved when using a Ct threshold for positivity of <49. However, detection sensitivity of *Pow* as the minor species was compromised once the ratio reached 1:10.

Based on the performance of the qPCR assays in mono- and mixed infections, an algorithm was developed to prevent false-positive detection of a secondary species within a mixed *Poc/Pow* infection. We found that this is more likely to happen if the majority species is abundant (Ct <39), leading to a cross-reactive false positive result for the other species that is indistinguishable from a small concentration of the minor species (Ct >44) (see data in **Table 2**). The resulting proposed classification system (**Fig 2**) does not indicate the relative abundance of the two species, but simply whether both are present. This approach maintains 100% specificity for both species based on the simulated mono-infections in Table 2 (0/55 and 0/45 false positives for *Poc* and *Pow*, respectively, as we did not include the $10^{-1}$ plasmid copies/$\mu$L runs for *Pow* given the limit of detection of 4.5 plasmid copies/$\mu$L for the *Pow* assay). Based on the mock mixture data in Fig 1, the modified qPCR assays paired with the proposed classification system achieve 84% sensitivity (280/335) for *Poc* (down to $10^{-1}$ plasmid copies/$\mu$L), and 94% sensitivity (239/255) for *Pow* (down to $10^0$ plasmid copies/$\mu$L).

**A**

| Mock plasmid mixtures (1:1) No. positive qPCR runs (mean Ct) | | |
| --- | --- | --- |
| | curtisi = wallikeri | |
| 18S plasmid copies/$\mu$L | Poc | Pow |
| | 1:1 | |
| $10^2$ | 10/10 (38.1) | 10/10 (37.8) |
| $10^1$ | 10/10 (43.1) | 10/10 (41.7) |
| $10^0$ | 9/10 (46.4) | 10/10 (45.4) |
| $10^{-1}$ | 3/10 (47.6) | 5/10 (46.4) |

**B**

| Mock plasmid mixtures of differing ratios No. positive qPCR runs (mean Ct) | | | | |
| --- | --- | --- | --- | --- |
| | curtisi > wallikeri | | curtisi < wallikeri | |
| 18S plasmid copies/$\mu$L | Poc | Pow | Poc | Pow |
| | 2:1 | | 1:2 | |
| $10^2$ | 10/10 (35.8) | 10/10 (39.0) | 10/10 (39.6) | 10/10 (40.1) |
| $10^1$ | 10/10 (39.9) | 10/10 (43.0) | 10/10 (42.7) | 10/10 (43.4) |
| $10^0$ | 10/10 (43.2) | 10/10 (45.7) | 7/10 (46.6) | 9/10 (47.1) |
| $10^{-1}$ | 4/10 (44.6) | 4/10 (46.0) | 2/10 (47.9) | 1/10 (48.0) |
| | 5:1 | | 1:5 | |
| $10^2$ | 10/10 (36.9) | 10/10 (40.9) | 10/10 (38.1) | 10/10 (38.5) |
| $10^1$ | 10/10 (41.0) | 10/10 (44.6) | 10/10 (42.2) | 10/10 (42.4) |
| $10^0$ | 10/10 (45.6) | 6/10 (47.5) | 3/10 (46.5) | 2/10 (47.0) |
| $10^{-1}$ | 3/10 (46.0) | 0/10 | 3/10 (46.5) | 2/10 (47.0) |
| | 10:1 | | 1:10 | |
| $10^2$ | 10/10 (34.5) | 10/10 (42.4) | 10/10 (39.3) | 10/10 (36.9) |
| $10^1$ | 10/10 (38.9) | 7/10 (45.1) | 10/10 (43.8) | 10/10 (40.8) |
| $10^0$ | 10/10 (43.3) | 2/10 (48.6) | 8/10 (46.5) | 10/10 (45.0) |
| $10^{-1}$ | 10/10 (46.5) | 0/10 | 2/10 (48.2) | 6/10 (47.7) |

**Fig 1. Performance of species-specific real-time PCR detection in mock mixtures of *P. ovale curtisi* and *wallikeri* 18S plasmid controls.** Mixtures were created either in equal proportions (A) or at ratios of 1:2, 1:5, and 1:10 (B). The number of positive runs and mean Ct of the positive qPCR runs for each species-specific assay is shown for 18S plasmid concentrations ranging from $10^2$ to $10^{-1}$ copies/$\mu$L. A Ct value <49 was used to determine a positive run.

## Performance of real-time PCR in clinical samples

Thirty-seven clinical blood samples from Cameroon and Tanzania that previously tested positive for *P. ovale* using a pan-ovale species 18S rRNA real-time PCR [26] were used to compare the performance of a published nested PCR [17] and the adapted real-time PCR (qPCR)

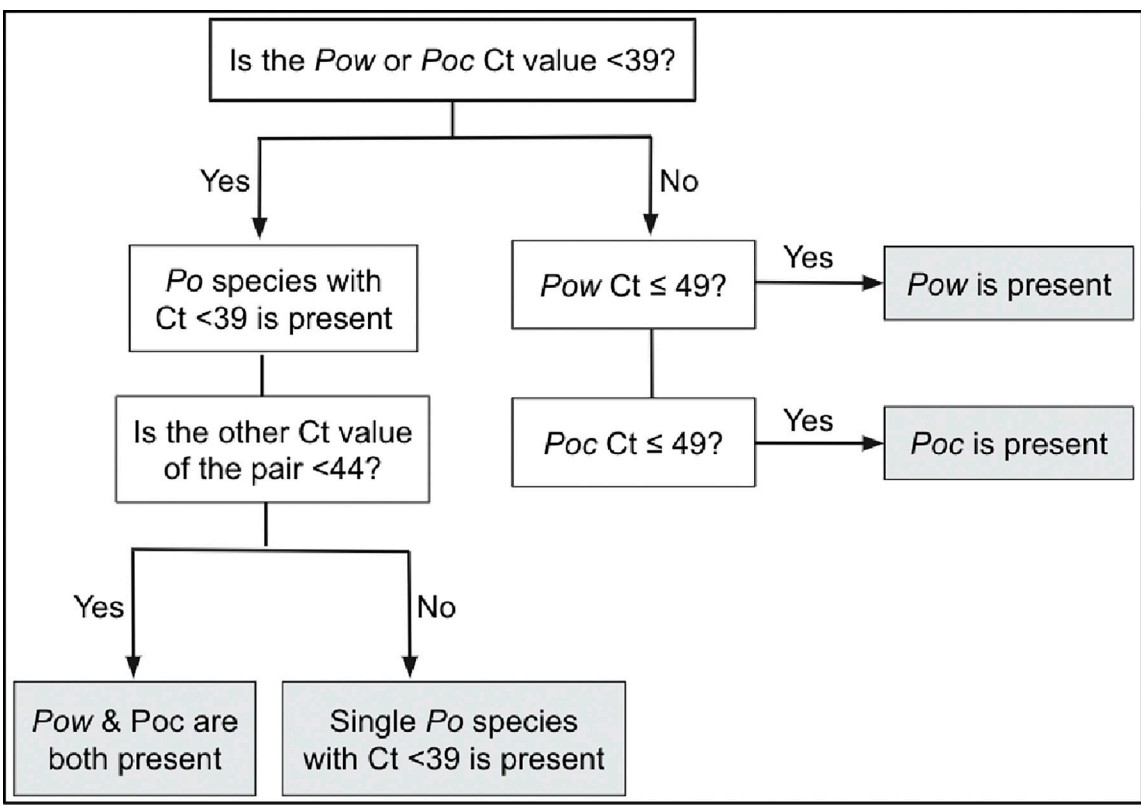

**Fig 2. Algorithm to determine the species of *P. ovale* malaria using 18S real-time PCR, whether as a single or mixed species infection.**

classification algorithm. Results were successfully obtained for 70% (26/37) of samples using qPCR versus 35% (13/37) of samples using published nested PCR conditions. When nested PCR was repeated with a greater number of cycles (up to 85 cycles across two rounds) and/or increased template DNA volume (up to 10 μL from 5 μL), 85% (23/27) could be successfully amplified (**Fig 3** and **Table 3**). Identified species was confirmed by Sanger sequencing of nPCR products obtained from leukodepleted blood (LDB) samples.

Good concordance was observed among the 19 samples successfully amplified in both nested PCR and real-time PCR assays. There was 100% agreement with regard to identification of the major *P. ovale* species present. However, real-time PCR additionally detected the presence of a second minor species in 2/19 samples that were not identified by nPCR (**Table 3**). Overall, among 28 unique *P. ovale*-infected individuals with blood samples with species determined by real-time PCR or nested PCR listed in Table 3, 54% (15/28) were infected with *Poc*, 36% (10/28) were infected with *Pow*, and 11% (3/28) were identified as harboring mixed *Poc*/*Pow* species infections.

Mosquito-based xenodiagnosis revealed a much higher prevalence of mixed *Poc*/*Pow* infections than that identified from human clinical blood samples. Seventeen oocyst-positive mosquito midguts obtained from mosquito feeds performed on 9 *P. ovale* infected persons in Tanzania, that had previously tested qPCR-positive for *P. ovale*, were available for species analysis. Fourteen of 17 (82%) midgut DNA samples successfully amplified in the *P. ovale* species qPCR assays, of which 11/14 (79%) were positive for both *Poc* and *Pow* using the classification scheme outlined in Fig 2 (**Table 4**). Of 9 *P. ovale* carriers, only 2 had mixed *P. ovale* infections that could be detected in the blood at the time of mosquito feeding, but mosquito-based

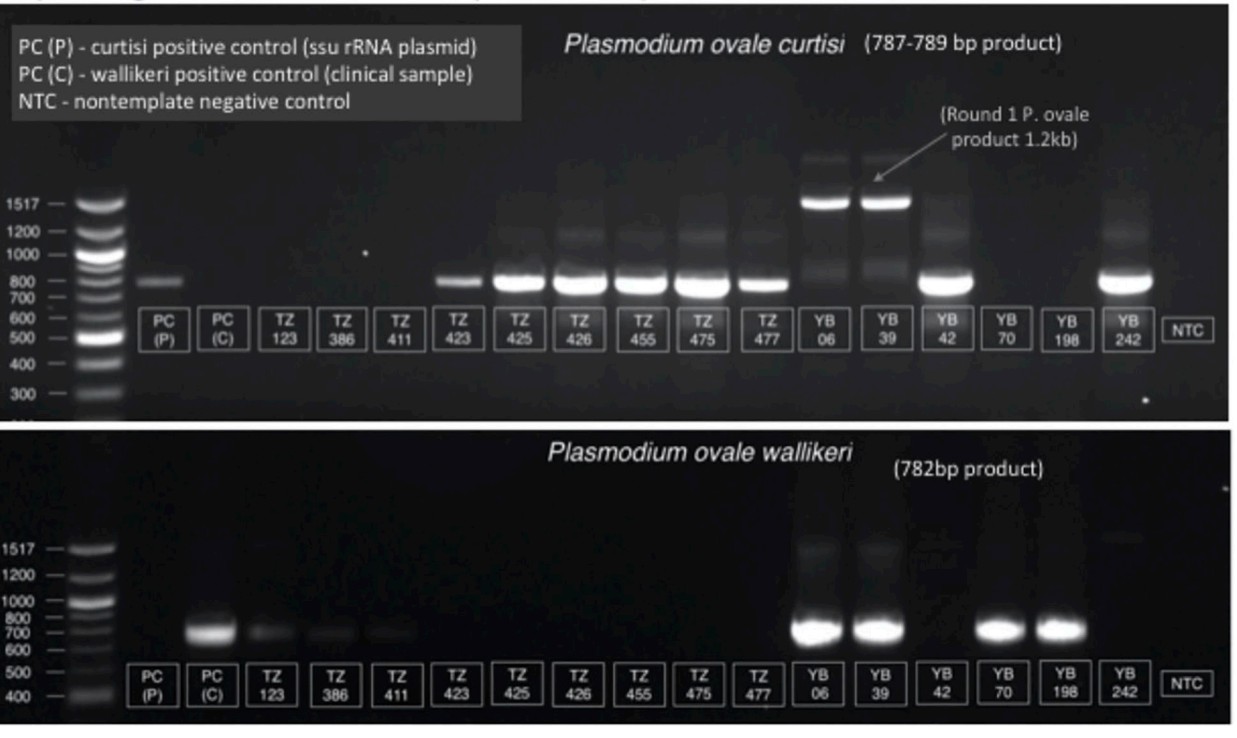

**Fig 3. Gel electrophoresis showing nested PCR results for select samples from Table 3.** The second round products of the nested PCR assay were run on a 1% agarose gel for *Poc* (top) and *Pow* (bottom) detection. Fragment lengths of 787–789 bp and 782 bp were expected for *Poc* and *Pow*, respectively. The results of the nested PCR assay on these human clinical blood samples, determined by the presence of a band on the depicted gel, are displayed in Table 3.

xenodiagnosis revealed that all but one (8/9) harbored both *Poc* and *Pow* and transmitted both species to mosquitoes. (**Table 4**).

## Discussion

By modifying *Poc*- and *Pow*-specific 18S rRNA real-time PCR assays and developing a classification algorithm to detect mixed *Poc/Pow* infections that avoids false-positive detection due to cross-reactivity, we show that mixed *Poc/Pow* infections occur naturally (~11% in our initial blood survey) and may be much more common than anticipated (89% by mosquito-based xenodiagnosis in our small sample). Given their sympatric distribution, co-transmission of both *Poc* and *Pow* species within the same *Anopheles* mosquitoes is not unexpected. Frequent co-transmission means that the two species have ample opportunity to recombine within mosquitoes. Yet a species barrier appears to be firmly established, likely due to prior distinct evolutionary pathways before their present co-existence in human hosts [1,2,9].

The higher frequency of *Poc/Pow* mixed infections we discovered among *Po*-infected mosquitoes compared to blood infection was a surprise. Xenodiagnosis has previously been suggested to be the most sensitive method for detecting human blood-stage infection, often detecting subpatent infections [32–34] and, in our experience, sometimes detecting parasitemias circulating just below the limit of detection of PCR [35]. Genetic diversity undetected by blood sampling can be revealed through mosquito sampling [36–39] and attests to the sampling efficiency of mosquitoes and transmission efficiency of gametocytes. While our results could be explained by a simultaneous outbreak of both *Po* species, the mosquito feeding assays depicted in Table 4 spanned two years of data collection. Rather, those exposed to one *Po*

**Table 3. Comparison of *P. ovale* species nested PCR and real-time PCR assays in field samples from Cameroon and Tanzania.** *Poc = P.o. curtisi; Pow = P.o. wallikeri.* Samples not successfully amplified in species assays are indicated with –.

| Sample ID | Po 18S (Ct value) | Nested PCR Po species | Real-time PCR (Ct value) P.o. curtisi | P.o. wallikeri | Po species | Concordant | Discordant |
|---|---|---|---|---|---|---|---|
| *Cameroon Samples (dried blood spots)* | | | | | | | |
| DSG2020_151 | 27.1 | Pow | 40.2 | 33.8 | Pow + Poc | | X |
| DSG2020_267 | 28.9 | Poc | 34.3 | – | Poc | X | |
| DSG2020_098 | 29.4 | Poc | 42.7 | 35.3 | Pow + Poc | | X |
| DSG2020_272 | 32.0 | Poc | 37.0 | 50.0 | Poc | X | |
| DSG2020_112 | 32.7 | Poc | 38.7 | – | Poc | X | |
| DSG2020_319 | 35.6 | – | 40.8 | – | Poc | | |
| DSG2020_097 | 36.0 | – | 49.9 | 42.8 | Pow | | |
| DSG2020_255 | 36.9 | Poc | 43.1 | – | Poc | X | |
| DSG2020_125 | 36.9 | – | 47.5 | 45.6 | Pow + Poc | | |
| DSG2020_138 | 40.0 | – | 47.0 | – | Poc | | |
| DSG2020_100 | 40.1 | – | – | 49.2 | – | | |
| DSG2020_101 | 40.3 | – | – | 47.1 | Pow | | |
| DSG2020_103 | 40.7 | Pow | – | – | – | | |
| DSG2020_096 | 41.7 | – | – | 49.8 | – | | |
| *Tanzania Samples (dried blood spots)* | | | | | | | |
| YB5006-MG[a] | 31.9 | Pow | 44.5 | 36.7 | Pow | X | |
| MqTZ-SN0423[a] | 32.7 | Poc | 44.9 | – | Poc | X | |
| MqTZ-SN0123[a] | 33.0 | Pow | – | 43.8 | Pow | X | |
| YB5042-MG[a] | 35.4 | Poc | 44.3 | – | Poc | X | |
| MqTZ-SN0386[a] | 35.9 | Pow | – | 48.5 | Pow | X | |
| YB5242-MG[a] | 37.2 | Poc | 45.1 | – | Poc | X | |
| MqTZ-SN0124[a] | 37.9 | – | 49.6 | x | – | | |
| YB5070-MG[a] | 40.6 | Pow | – | 47.6 | Pow | X | |
| YB5198-MG[a] | 41.7 | Pow | – | – | – | | |
| MqTZ-SN0411[a] | 41.9 | Pow | – | – | – | | |
| *Tanzania Samples (leukodepleted blood)* | | | | | | | |
| MqTZ-0477[b] | 33.7 | Poc | 38.7 | 46.4 | Poc | X | |
| MqTZ-0123[b] | 33.8 | Pow | – | 43.8 | Pow | X | |
| MqTZ-0426[b] | 34.0 | Poc | 40.0 | – | Poc | X | |
| MqTZ-0475[b] | 35.6 | Poc | 41.1 | – | Poc | X | |
| MqTZ-0425[b] | 40.1 | Poc | 47.3 | – | Poc | X | |
| MqTZ-0455[b] | 42.1 | Poc | 48.5 | – | Poc | X | |
| MqTZ-2394 | 27.7 | – | 44.5 | 37.4 | Pow | | |
| MqTZ-2499 D14 | 29.8 | – | 37.3 | – | Poc | | |

[a]Nested PCR required increase in amplification cycles and/or increased volume of template DNA

[b]Confirmed by Sanger sequencing of nPCR product

Note: Five samples did not amplify in either the nested PCR or real-time PCR assay.

species may be more likely to also be exposed to the other *Po* species, with malaria exposure concentrated in a small proportion of the population who then serve as a reservoir [40]. The role of relapse, in which persons once exposed remain latently infected in the liver, could also contribute to unexpected high rates of co-infection. *Po* species inoculated separately could relapse in unison when conditions are right. Larger molecular studies in other settings and including wild-caught mosquitoes are needed to verify our findings.

**Table 4. Real-time qPCR detection of *P. ovale* species within *P. ovale*-infected mosquito midgut samples from Tanzania.** *Poc = P.o. curtisi; Pow = P.o. wallikeri.*

| Sample ID | Blood | | | | Mosquito midgut | | | | |
|---|---|---|---|---|---|---|---|---|---|
| | Po 18S Ct (blood)* | *P.o. curtisi Ct* | *P.o. wallikeri Ct* | Po species identified | Midgut ID | Po 18S Ct (midgut)** | *P.o. curtisi Ct* | *P.o. wallikeri Ct* | P.o. species identified |
| MqTZ-1610 | 32.2 | 34.8 | N/A | *Poc* | L M2 | 13.6 | 39.3 | 43.3 | *Poc + Pow* |
| | | | | | N2 M1 | 21.1 | 41.8 | 40.7 | *Pow + Poc* |
| | | | | | R M5 | 21.9 | N/A | N/A | – |
| | | | | | N1 M4 | 27.8 | N/A | N/A | – |
| MqTZ-3863 | 32.3 | 41.7 | 38.2 | *Pow + Poc* | R M9 | 19.5 | 35.4 | 34.0 | *Pow + Poc* |
| | | | | | R M5 | 21.4 | 35.4 | 34.0 | *Pow + Poc* |
| MqTZ-3059 | 37.5 | 47.8 | 44.0 | *Pow + Poc* | R M1 | 19.7 | 43.2 | 44.4 | *Poc + Pow* |
| MqTZ-191 | 39.9 | – | – | – | A2 M17 | 19.1 | 40.5 | 35.1 | *Pow + Poc* |
| MqTZ-455 | 42.1 | 47.4 | N/A | *Poc* | L M15 | 18.5 | 41.0 | 47.4 | *Poc + Pow* |
| MqTZ-167 | 42.8 | – | – | – | L M1 | 16.0 | 31.1 | 30.6 | *Pow + Poc* |
| | | | | | L M8 | 20.7 | 35.7 | N/A | *Poc* |
| MqTZ-1077 | 44.3 | 49.5 | N/A | – | L M1wk4 | 19.1 | 31.2 | N/A | *Poc* |
| MqTZ-1733 | 43.8 | 48.9 | N/A | *Poc* | R M1 | 20.4 | 40.6 | 44.0 | *Poc + Pow* |
| | | | | | L M1 | 28.1 | N/A | N/A | – |
| MqTZ-799 | 44.7 | – | – | – | R M1 | 23.2 | 46.7 | 45.5 | *Pow + Poc* |
| | | | | | L M1 | 24.2 | 46.5 | N/A | *Poc* |
| | | | | | L M2 | 28.7 | 45.7 | 48.1 | *Poc + Pow* |

*Cts reported are from leukodepleted blood samples collected at the time of mosquito feeding, except for MqTZ-191, in which DNA was extracted from a dried blood spot was used.

**Mosquito midgut qPCRs were performed on a 1:50 dilution of a conventional genus-level "round 1" PCR product, run out to 40 cycles

This work is not without limitations. First, our experiments to determine analytical sensitivity and develop a species classification scheme used plasmid controls, and we did not specifically test their robustness within blood or mosquito samples. Our finding that co-infection by both species was more common in mosquitoes than in human blood samples was surprising and needs to be validated with further studies. However, we expect lower assay sensitivity in mosquito samples due to low parasite burdens and to PCR inhibition rather than enhanced cross-reactivity or compromised diagnostic specificity. Second, our classification scheme is expected to underestimate mixed *Poc/Pow* infections when the majority species is at high density. Third, given the slightly greater sensitivity of the *Poc* real-time PCR assay over the *Pow* assay, we cannot draw firm conclusions about the relative prevalence of *Poc* and *Pow* in our surveys, aside from finding that both were represented in blood and mosquito samples. Fourth, though we showed excellent concordance of the real-time PCR assays with nested PCR and Sanger sequencing, we did not attempt to sequence our predominantly-mixed midgut samples to verify the presence of both species due to the paucity of DNA available and the bioinformatic challenge of detecting mixed *Poc/Pow* infections by targeted sequencing [36–26]. Finally, some of our midgut PCR results involved high Ct values and might reflect contamination of samples, but the three-year time span of sample collection and the detection of *Poc* single species midguts (as well as midguts that did not amplify) make this less likely.

In conclusion, the real-time PCR approach described here represents an efficient method for detecting mixed *Poc/Pow* infections in both human clinical blood samples and mosquito midguts. Mixed *Poc/Pow* infections were commonly detected in mosquito midguts, and were

also detected, albeit to a lesser degree, in both human dried blood spots and leukocyte-depleted blood samples. This suggests that the extent of mixed *Poc/Pow* infection may be greater than previously appreciated. Issues with cross-reactivity remain with the real-time PCR assays, which would best be resolved by using separate species-specific gene targets that would allow development of primer and probe sets with no potential for cross-reactivity [41]. Much remains to be learned about *Poc* and *Pow* epidemiology in sub-Saharan Africa, including how they may be evolving in the face of malaria control efforts designed to target *P. falciparum*. Recognition of the limitation of current assays for detecting co-infection and continued development of better, more facile diagnostics will improve our ability to understand whether and how these two species potentially differ in their epidemiology, biology, and clinical manifestations. Our findings suggest that the degree to which these closely related but sympatric species co-circulate within their human and mosquito hosts may be underappreciated.

## Acknowledgments

We thank the study participants, as well as community partners and research staff that carried out the field collections in Tanzania and Cameroon.

## Author Contributions

**Conceptualization:** Jonathan J. Juliano, Jonathan B. Parr, Jessica T. Lin.

**Formal analysis:** Feng-Chang Lin.

**Funding acquisition:** Jonathan B. Parr, Jessica T. Lin.

**Investigation:** Varun R. Potlapalli, Meredith S. Muller, Billy Ngasala, Innocent Mbulli Ali, Yu Bin Na, Danielle R. Williams, Srijana Chhetri, Mei S. Liu, Derrick Mathias, Brian B. Tarimo.

**Methodology:** Varun R. Potlapalli, Meredith S. Muller, Yu Bin Na, Danielle R. Williams, Oksana Kharabora, Srijana Chhetri, Derrick Mathias, Jessica T. Lin.

**Project administration:** Meredith S. Muller, Jonathan J. Juliano, Jonathan B. Parr, Jessica T. Lin.

**Resources:** Billy Ngasala, Innocent Mbulli Ali, Brian B. Tarimo, Jessica T. Lin.

**Supervision:** Oksana Kharabora, Jessica T. Lin.

**Validation:** Varun R. Potlapalli, Meredith S. Muller, Yu Bin Na, Danielle R. Williams, Srijana Chhetri.

**Visualization:** Varun R. Potlapalli, Meredith S. Muller, Yu Bin Na, Jessica T. Lin.

**Writing – original draft:** Varun R. Potlapalli, Meredith S. Muller, Jessica T. Lin.

**Writing – review & editing:** Varun R. Potlapalli, Innocent Mbulli Ali, Kelly Carey-Ewend, Jonathan J. Juliano, Jonathan B. Parr, Jessica T. Lin.

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
