## [Decision Letter · Decision Letter 0]

3 Jul 2023

Dear Dr. Lin,

Thank you very much for submitting your manuscript "Real-time PCR detection of mixed Plasmodium ovale curtisi and wallikeri species infections in human and mosquito hosts" for consideration at PLOS Neglected Tropical Diseases. As with all papers reviewed by the journal, your manuscript was reviewed by members of the editorial board and by several independent reviewers. In light of the reviews (below this email), we would like to invite the resubmission of a significantly-revised version that takes into account the reviewers' comments. 

We cannot make any decision about publication until we have seen the revised manuscript and your response to the reviewers' comments. Your revised manuscript is also likely to be sent to reviewers for further evaluation.

Sincerely,

Paul O. Mireji, PhD

Academic Editor

Charles Jaffe

Section Editor

Reviewer's Responses to Questions

**Key Review Criteria Required for Acceptance?**

**Methods**

-Are the objectives of the study clearly articulated with a clear testable hypothesis stated?

-Is the study design appropriate to address the stated objectives?

-Is the population clearly described and appropriate for the hypothesis being tested?

-Is the sample size sufficient to ensure adequate power to address the hypothesis being tested?

-Were correct statistical analysis used to support conclusions?

-Are there concerns about ethical or regulatory requirements being met?

Reviewer #1: see below

Reviewer #2: The authors have introduced an optimized protocol that effectively discriminates between Plasmodium ovale curtisi (Poc) and Plasmodium ovale wallikeri (Pow) in mixed P. ovale species infections. However, there are certain aspects regarding the description of the population (sampling) used for protocol optimization that require clarification. Notably, the qualitative and quantitative details of the study population are not explicitly provided. For instance, in the discussion of the modified PCR on lines 114 and 119, the sample size remains unclear. In order to enhance the methodology section, it is imperative to present a more coherent and sequential flow of the protocol, rather than a fragmented description of the modifications, accompanied by appropriate statistical treatment. Addressing these concerns will greatly enhance the transparency and reproducibility of the study.

Reviewer #3: The methods should give more details on equipment. What real-time PCR machine was used?

**Results**

-Does the analysis presented match the analysis plan?

-Are the results clearly and completely presented?

-Are the figures (Tables, Images) of sufficient quality for clarity?

Reviewer #1: see below

Reviewer #2: The methodology for determining the sensitivity and specificity values presented in lines 183-184 requires further clarification. It is not apparent how these values were derived, and a detailed explanation of true and false positives would enhance the transparency of the data provided. Additionally, it would be beneficial to describe the criteria used for establishing the Ct value threshold. Elaborating on these aspects will provide a clearer understanding of the methodology employed and strengthen the interpretation of the results.

Reviewer #3: Regarding “…detected down to 100 18S plasmid copies/μL of their respective P. ovale species, or the equivalent of 1-2 parasites/μL assuming 5-8 copies of 18S rRNA per genome”: Why not just say ‘one’? (100 = 1) So this is saying that 1 plasmid copy is equivalent to 1-2 parasites assuming 5-8 copies per genome. It does not quite make sense. Unless it was supposed to be 101.

Ct < 49 is still a pretty high threshold for positivity. Melting or dissociation curve analysis would have helped to confirm that amplicons are of target sequences.

Though I understand the algorithm conceptually, it seems to be based on somewhat arbitrary thresholds. Further I fear that it is based on the cross-activity first observed at 103 (or 1000) copies. As the next previous dilution tested where no cross-activity was observed was at 102 (100) copies, the threshold of cross-reactivity is not clear. It could be 200 copies… The observed mixed infection in this study, especially of the surprising mosquito experiment results, should be confirmed with amplicon sequencing.

**Conclusions**

-Are the conclusions supported by the data presented?

-Are the limitations of analysis clearly described?

-Do the authors discuss how these data can be helpful to advance our understanding of the topic under study?

-Is public health relevance addressed?

Reviewer #1: see below

Reviewer #2: (No Response)

Reviewer #3: (No Response)

**Editorial and Data Presentation Modifications?**

Reviewer #1: (No Response)

Reviewer #2: (No Response)

Reviewer #3: The last two sentences of the introduction are results and discussion and should not be part of the introduction. The introduction should simply give background and rationale for the study and state the objectives.

The beginning of the discussion should be part of the introduction.

If at the beginning of a sentence, genus names should not be abbreviated, even if previously defined.

I don’t think that Tables should be formatted like Table 1 with all these paragraphs within cells. And the entire methods section should not be in landscape.

**Summary and General Comments**

Reviewer #1: The authors describe the detection of Plasmodium ovale wallikeri and P. o. curtisi in humans and mosquitoes using a real-time PCR. The assay is of high interest and might allow specification at low parasitemia/DNA amounts. The quality of the manuscript is very good. 

Some modifications are recommended:

Title – species can be deleted; it is recommended to change to … Plasmodium ovale curtisi and P. ovale wallikeri infections…

Line 21 – malaria species is not correct. Malaria is a disease; pathogen/parasite/Plasmodium species cause diseases/infections

Methods – Please provide primer names and sequences (and bp sizes of PCR products)

Methods/Discussion – Table 1 - There are several other protocols to discriminate PoC/PoW (e.g. Fuehrer et al. JCM; Lamien-Meda et al. Acta Tropica). Please check literature.

Reviewer #2: (No Response)

Reviewer #3: Though I can see considerable work gone into this study, I must recommend a major revision to clarify the significance of the study and to validate some of the confusing findings.

It would be helpful if the paper would early on and in the abstract/summary say why it is important to identify mixed Po infections. It is not clear to me. The introduction should have more background on why it is important to identify mixed Poc and Pow infections. Is there a reason that this could contribute to disease or understanding of epidemiology somehow?

PLOS authors have the option to publish the peer review history of their article (what does this mean?). If published, this will include your full peer review and any attached files.

Reviewer #1: Yes: Hans-Peter Fuehrer

Reviewer #2: No

Reviewer #3: No
---

## [Decision Letter · Decision Letter 1]

17 Oct 2023

Dear Dr. Lin,

Thank you very much for submitting your manuscript "Real-time PCR detection of mixed Plasmodium ovale curtisi and wallikeri infections in human and mosquito hosts" for consideration at PLOS Neglected Tropical Diseases. As with all papers reviewed by the journal, your manuscript was reviewed by members of the editorial board and by several independent reviewers. The reviewers appreciated the attention to an important topic. Based on the reviews, we are likely to accept this manuscript for publication, providing that you modify the manuscript according to the review recommendations. 

From our editorial perspective, we think the manuscript will be if the following aspects are addressed 

1) At the end of the introduction, the authors discusses results that have not yet been presented and conclusions appears to be "prematurely" drawn.

"Our optimized protocol for running parallel Poc and Pow species-specific qPCRs detected mixed Po species infection in a smaller proportion of blood samples from Tanzania and Cameroon. However, this percentage was much higher in Po-positive mosquito midgut samples derived from mosquito feeding studies in Tanzania, raising the possibility that mixed Poc/Pow infections may be more common than currently appreciated."

2)I|n the discussion section, the information at the beginning fits better ( strengthen the introduction) than with the discussion. Can the authors consider this suggestion? 

"Improved detection and rising prevalence of non-falciparum malaria species in sub-Saharan Africa [4–6,8] has spurred interest in better defining their epidemiology and biology [3,9,32,33]. Now recognized as two separate species, Plasmodium ovale curtisi (Poc) and Plasmodium ovale wallikeri (Pow) appear to circulate sympatrically, yet it appears <20 cases of Poc/Pow co-infection have been reported across >35 studies encompassing 1,515 P. ovale cases in the literature [3,13,15,16,22,26]."

Sincerely,

Paul O. Mireji, PhD

Academic Editor

Charles Jaffe

Section Editor

From our editorial perspective, we think the manuscript will be if the following aspects are addressed 

1) At the end of the introduction, the authors discusses results that have not yet been presented and conclusions appears to be "prematurely" drawn.

"Our optimized protocol for running parallel Poc and Pow species-specific qPCRs detected mixed Po species infection in a smaller proportion of blood samples from Tanzania and Cameroon. However, this percentage was much higher in Po-positive mosquito midgut samples derived from mosquito feeding studies in Tanzania, raising the possibility that mixed Poc/Pow infections may be more common than currently appreciated."

2)I|n the discussion section, the information at the beginning fits better ( strengthen the introduction) than with the discussion. Can the authors consider this suggestion? 

"Improved detection and rising prevalence of non-falciparum malaria species in sub-Saharan Africa [4–6,8] has spurred interest in better defining their epidemiology and biology [3,9,32,33]. Now recognized as two separate species, Plasmodium ovale curtisi (Poc) and Plasmodium ovale wallikeri (Pow) appear to circulate sympatrically, yet it appears <20 cases of Poc/Pow co-infection have been reported across >35 studies encompassing 1,515 P. ovale cases in the literature [3,13,15,16,22,26]."

Reviewer's Responses to Questions

**Key Review Criteria Required for Acceptance?**

**Methods**

-Are the objectives of the study clearly articulated with a clear testable hypothesis stated?

-Is the study design appropriate to address the stated objectives?

-Is the population clearly described and appropriate for the hypothesis being tested?

-Is the sample size sufficient to ensure adequate power to address the hypothesis being tested?

-Were correct statistical analysis used to support conclusions?

-Are there concerns about ethical or regulatory requirements being met?

Reviewer #2: (No Response)

Reviewer #3: (No Response)

**Results**

-Does the analysis presented match the analysis plan?

-Are the results clearly and completely presented?

-Are the figures (Tables, Images) of sufficient quality for clarity?

Reviewer #2: (No Response)

Reviewer #3: (No Response)

**Conclusions**

-Are the conclusions supported by the data presented?

-Are the limitations of analysis clearly described?

-Do the authors discuss how these data can be helpful to advance our understanding of the topic under study?

-Is public health relevance addressed?

Reviewer #2: (No Response)

Reviewer #3: (No Response)

**Editorial and Data Presentation Modifications?**

Reviewer #2: (No Response)

Reviewer #3: (No Response)

**Summary and General Comments**

Reviewer #2: This is a much improved version of the MS. The concerns have been sufficiently addressed.

Reviewer #3: (No Response)

PLOS authors have the option to publish the peer review history of their article (what does this mean?). If published, this will include your full peer review and any attached files.

Reviewer #2: No

Reviewer #3: No

Figure Files:

Data Requirements:

Reproducibility:

References

---

## [Editor Report · Decision Letter 2]

21 Nov 2023

Dear Dr. Lin,

We are pleased to inform you that your manuscript 'Real-time PCR detection of mixed Plasmodium ovale curtisi and wallikeri infections in human and mosquito hosts' has been provisionally accepted for publication in PLOS Neglected Tropical Diseases.

Best regards,

Charles L. Jaffe, Ph.D.

Section Editor

Charles Jaffe

Section Editor

---

## [Editor Report · Acceptance letter]

30 Nov 2023

Dear Dr. Lin,

We are delighted to inform you that your manuscript, "Real-time PCR detection of mixed *Plasmodium ovale curtisi* and *wallikeri* infections in human and mosquito hosts," has been formally accepted for publication in PLOS Neglected Tropical Diseases.

Best regards,

Shaden Kamhawi

co-Editor-in-Chief

Paul Brindley

co-Editor-in-Chief
